# Protective role of Pannexin1 in lymphatic endothelial cells in the progression of atherosclerosis in female mice

Avigail Ehrlich[1,2], Graziano Pelli[1,2], Bernard Foglia[1,2], Filippo Molica[1,2☯], Brenda R. Kwak [1,2☯] *

1 Department of Pathology and Immunology, University of Geneva, Geneva, Switzerland, 2 Geneva Center for Inflammation Research, Faculty of Medicine, University of Geneva, Geneva, Switzerland

☯ These authors contributed equally to this work.
* Brenda.KwakChanson@unige.ch

**Data Availability Statement:** The source data that support the findings will be available in YARETA following acceptance of the manuscript.

## Abstract

Atherosclerosis is a progressive arterial disease arising from imbalanced lipid metabolism and a maladaptive immune response. The lymphatic system ensures tissue fluid homeostasis, absorption of dietary fats and trafficking of immune cells to draining lymph nodes, thereby potentially affecting atherogenesis. Endothelial cell-specific deletion of Pannexin1 (Panx1) in apolipoprotein E-deficient ($Apoe^{-/-}$) mice increased atherosclerosis, suggesting a protective role for Panx1 channels in arterial endothelial function. Here, we investigated the role of Panx1 in lymphatic endothelial cells (LECs) in the initiation and the progression of atherosclerosis. Male or female $Prox1-CreER^{T2+}Panx1^{fl/fl}Apoe^{-/-}$ and $Panx1^{fl/fl}Apoe^{-/-}$ mice were fed a high cholesterol diet (HCD) for 6 or 10 weeks. Tamoxifen-induced deletion of Panx1 was performed before or after 4 weeks of HCD. Body weight and serum lipid profiles were determined. The atherosclerotic plaque burden was assessed by Sudan-IV staining on thoracic-abdominal aortas and in aortic roots. Plaque composition was determined by immunohistochemistry. No differences in serum cholesterol, LDL and HDL were observed between genotypes and between sexes after HCD. Bodyweight, serum triglycerides and free fatty acid levels were higher before and after 6 weeks of HCD in male $Prox1-CreER^{T2+}Panx1^{fl/fl}Apoe^{-/-}$ and control $Panx1^{fl/fl}Apoe^{-/-}$ mice compared to females of the same genotypes, which was associated with more lipids and inflammatory cells in their atherosclerotic plaques. In contrast, the atherosclerotic plaque burden was higher in female mice. The progression of atherosclerosis in male mice was not different between genotypes. However, female $Prox1-CreER^{T2+}Panx1^{fl/fl}Apoe^{-/-}$ mice showed enhanced progression of atherosclerosis compared to $Panx1^{fl/fl}Apoe^{-/-}$ controls of the same sex. In addition, atherosclerotic lesions in female, but not in male, $Prox1-CreER^{T2+}Panx1^{fl/fl}Apoe^{-/-}$ mice showed T cell enrichment. Altogether, our results reveal differential sex-dependent effects of Panx1 in lymphatic endothelium on the progression of atherosclerosis.

**Funding:** This study was supported by grants from the Swiss National Science Foundation (grant number 310030_182573 to BRK) and the Gabbiani fund (to AE). The funders had no role in study design, data collection and analysis, decision to publish, or preparation of the manuscript.

**Competing interests:** The authors have declared that no competing interests exist.

## Introduction

The lymphatic vasculature forms an unidirectional transport pathway from the interstitial space to the systemic veins. The lymphatic system comprises a network of blind-ended capillaries, valved collecting vessels, larger ducts and lymph nodes (LNs), which ensures tissue fluid homeostasis, absorption of dietary fats and trafficking of immune cells such as dendritic cells (DCs) to draining LNs [1]. Lymphangiogenesis, the formation of new lymphatic vessels from existing ones, occurs mainly in response to the activation of vascular endothelial growth factor receptor 3 (VEGFR3) by vascular endothelial growth factor C (VEGFC), which stimulates the proliferation of lymphatic endothelial cells (LECs) [2]. LECs are among the principal actors safeguarding the different functions of the lymphatic system [3]. Dysfunctional lymphatic vessels give rise to various pathological conditions, including inherited and acquired forms of lymphedema, malabsorption syndromes, autoimmune disorders and immune deficiencies [4]. Recent evidence points to a role for the lymphatic system in atherosclerosis [5, 6].

Atherosclerosis is the underlying cause of myocardial infarction and stroke and is the world's leading cause of death [7]. This progressive disease affecting medium-sized and large arteries is characterized by an endothelial dysfunction, accumulation of lipids in the intima and an inflammatory response leading to the formation of atherosclerotic plaques [8]. The progression of the disease implies further recruitment of inflammatory cells, smooth muscle cell (SMC) migration and proliferation, cell death and fibrosis. Together, these factors determine atherosclerotic plaque stability and the risk of plaque rupture leading to acute cardiovascular events [9]. Stable atherosclerotic lesions display a thick fibrous cap with a high content of SMCs, a small necrotic core and a low macrophage content [10].

Both in mice and human, lymphatic vessels are mostly present in the arterial adventitial layer, and arteries with a dense lymphatic network seem naturally protected against atherosclerosis [5, 11]. Two independent atherosclerosis-susceptible mouse models with severe lymphatic insufficiency showed increased serum cholesterol levels and enhanced atherogenesis, pointing to a protective role for lymphatic vessels in maintaining proper lipoprotein and vascular homeostasis [12]. In agreement, dissection of plaque draining LNs and lymphatic vessels in apolipoprotein E-deficient ($Apoe^{-/-}$) mice worsened atherosclerosis irrespective of serum lipids, with T lymphocyte accumulation in the atherosclerotic plaques and in the adventitia [13]. In addition, inhibition of VEGFR3-mediated signaling caused T cell enrichment in atherosclerotic lesions, without compromising adventitial lymphatic density, suggesting the involvement of VEGFR3-independent growth of lymphatics [13]. Interestingly, the membrane channel-forming protein pannexin1 (Panx1) is highly expressed in human LECs and has been shown to regulate lymphangiogenesis *in vitro* [14].

Panx1 channels allow for the diffusion of ions and signaling molecules between the cytosol and the extracellular environment [15, 16]. Panx1 channels control the inflammatory response by regulating inflammasome activation and leukocyte migration [15]. We have previously shown that atherosclerosis was enhanced in *Tie2-Cre*$^{Tg}$*Panx1*$^{fl/fl}$*Apoe*$^{-/-}$ mice, pointing to an atheroprotective role for Panx1 in endothelial and/or monocytic cells. Interestingly, *Panx1*$^{-/-}$*Apoe*$^{-/-}$ mice showed impairment of lymphatic function with reduced dietary fat absorption, which counterbalanced the proatherogenic effect of *Tie2*-Cre-induced Panx1 deletion from these cells [17].

Taken together, these studies suggest a central role for Panx1 in linking lymphatic function to lipid metabolism and atherogenesis. However, whether specific Panx1 deletion in LECs affects atherosclerotic plaque burden and composition remains yet to be discovered.

## Methods

### Animals

Animal experimentation conformed to the Guide for the Care and Use of Laboratory Animals published by the US National Institutes of Health (NIH Publication No. 85-23, revised 1966) and all experiments were approved by the Swiss cantonal veterinary authorities. All mice were housed in a conventional standard animal facility at 24°C with light cycles of 12 hours of light and 12 hours of night. Cages were enriched with a carton house and cottons. Male and female *Prox1CreER^{T2}Panx1^{fl/fl}Apoe^{-/-} (Panx1^{LECdel}Apoe^{-/-})* and *Panx1^{fl/fl}Apoe^{-/-}* control mice, on a C57BL/6J background, were used in this study (Fig 1). All mice received twice a day, for 4 consecutive days, intraperitoneal injections of 1 mg tamoxifen (Sigma) in kolliphor oil (Sigma), either before high cholesterol diet (HCD; Fig 1A) or after 4 weeks of HCD (Fig 1B). This protocol is known to induce effective and specific Panx1 deletion from LECs [18]. Submandibular blood sampling was performed under short-term anesthesia with isofluorane inhalation (4%). Mice were fed for a total of 6 or 10 weeks with a HCD (1.25% cholesterol, 0% cholate; Safe diets). Thereafter, mice were killed under general anesthesia induced by an intraperitoneal injection of ketarom (10 mg/kg xylazine mixed with 100 mg/kg ketamine).

### Blood and tissue collection

Blood was collected by cardiac puncture and serum was obtained after 15 min centrifugation at 5000 rpm at 4°C. The mice were then perfused with 0.9% NaCl and organs were dissected.

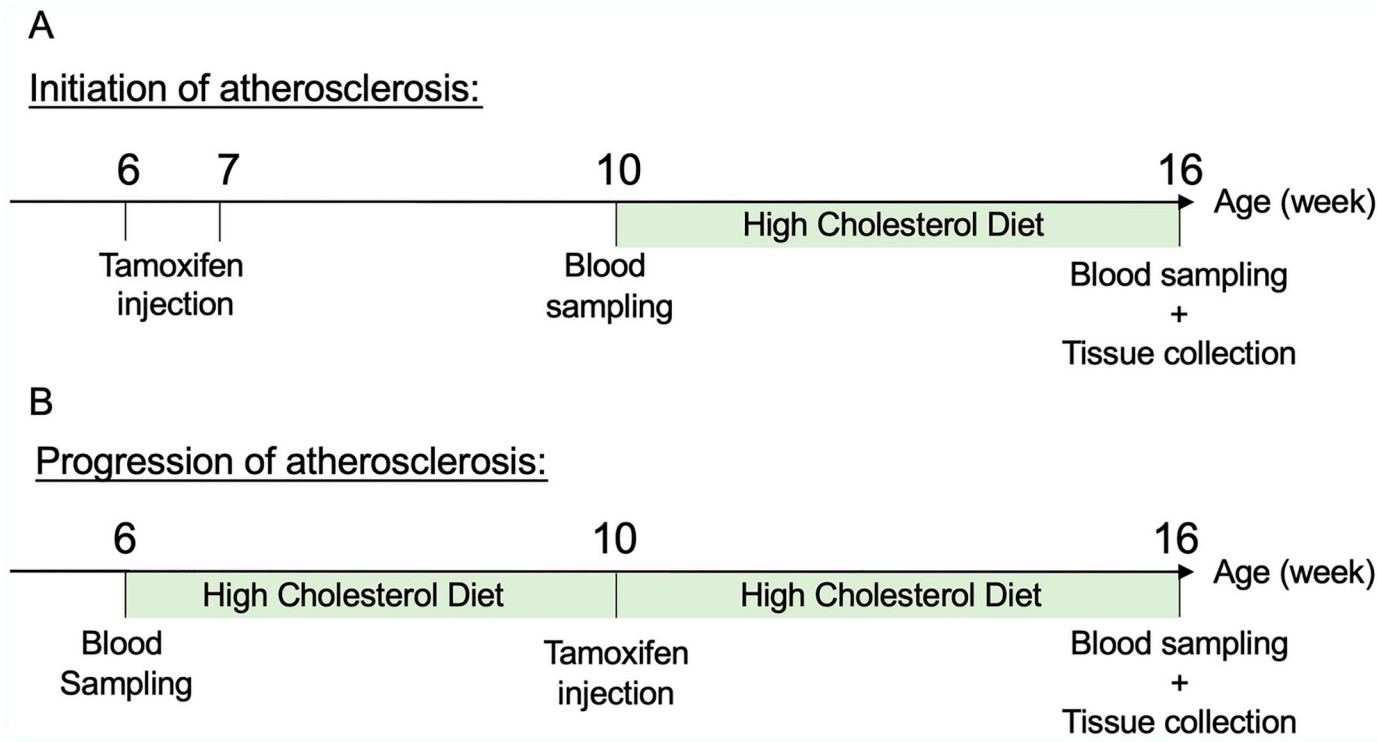

**Fig 1. Timeline for initiation and progression of atherosclerosis.** (A) For the initiation study, Panx1 deletion was induced at 6 weeks of age by tamoxifen injection for consecutive 4 days. When mice reached 10 weeks of age, blood was taken from female and male *Panx1^{LECdel}Apoe^{-/-}* and control *Panx1^{fl/fl}Apoe^{-/-}* mice and they were placed on a HCD for 6 weeks. At 16 weeks of age, mice were killed and blood and different tissues were collected. (B) For the progression study, blood was sampled at 6 weeks of age and atherosclerosis was induced in control *Panx1^{fl/fl}Apoe^{-/-}* and *Panx1^{LECdel}Apoe^{-/-}* mice by feeding them with a HCD for 10 weeks. Four weeks after the start of the HCD, Panx1 deletion was induced with tamoxifen injections. At the end of the 10 weeks HCD period, mice were killed and blood and tissues were collected.

Aortic roots were embedded in OCT compound (Tissue-Tek; Sakura) and snap-frozen. Thoracic-abdominal aortas were fixed in 4% paraformaldehyde (PFA) for 24 hours, rinsed with phosphate-buffered saline (PBS), incubated overnight in a Sudan-IV solution and longitudinally opened along the ventral midline, as described previously [19, 20]. The extent of atherosclerosis was determined by dividing Sudan-IV-positive area by the total surface of the thoracic-abdominal aorta. Atherosclerotic plaque burden was also analyzed in 5 serial cryosections of aortic roots (5 μm thickness, inter-distance 50 μm). Images were captured with a Zeiss Axioscan.Z1 automated slide scanner and quantification was performed by computer image analysis using Qpath v0.4.3 software. The extent of atherosclerosis was determined by dividing atherosclerotic area by the total surface of the aortic roots for each cryosection. An average value of all cryosections was calculated for each mouse.

## Histological analysis

Immunostaining on cryosections of aortic roots involved different protocols. For macrophages, the staining was performed using CD68 primary antibody (1:1000; Biorad). In brief, after fixation with acetone (5 min), neutralization with 0.5 mol/L $NH_4Cl$ (15 min), incubation for 10 min with BLOXALL (Vector Laboratories) and blocking with 2.5% normal horse serum (10 min), sections were incubated with the primary antibody at room temperature (RT) for 90 min. Staining was visualized using a secondary antibody coupled to streptavidin detected with a 3,3′-Diaminobenzidine (DAB) substrate kit (Vector Laboratories). Sections were counterstained with a hemalun Meyer's solution (Merck) and finally mounted with Aquatex (Sigma). For SMCs, the staining was performed using α-smooth muscle actin antibody (α-SMA; 1:50, kindly provided by M-L Bochaton-Piallat). In brief, after fixation with 4% PFA (15 min) and post-fixation/permeabilization at -20˚C with MeOH (3 min), sections were incubated for 1 hour at RT with the primary antibody, which was detected using a goat anti-mouse IgG2a Alexa 488 antibody (1:100; Jackson Laboratories) for 30 min at RT. Nuclei were stained with 4′,6-diamidino-2-phenylindole (DAPI; 1/20000; Invitrogen) for 10 min. Slides were mounted with Vectashield mounting medium (Vector Laboratories). For T lymphocytes, the staining was performed using CD3 primary antibody (1:100; CD3-12; Biorad). In brief, after fixation with PFA (15 min), permeabilization with 0.3% Triton-X100 (15 min), neutralization with 0.5 mol/L $NH_4Cl$ (15 min), and blocking with 10% normal goat serum (10 min; Vector Laboratories), sections were overnight incubated with the primary antibody at 4˚C. Staining was visualized using a goat anti-rat Alexa488 secondary antibody (1:100; Jackson Laboratories). Nuclei were stained with 4′,6-diamidino-2-phenylindole (DAPI; 1/20000; Invitrogen) for 10 min. Slides were mounted with Vectashield mounting medium (Vector Laboratories). Collagen was visualized and quantified after Masson's trichrome staining. Lymphatic vessels were fluorescently stained using a LYVE-1 antibody (1:500; Angiobio). In brief, after fixation with 4% PFA (15 min), permeabilization with 0.2% Triton X-100 during 1 hour and blocking for 30 min with 2% bovine serum albumin, sections were incubated overnight with the primary antibody. Sections were then incubated with an Alexa 568 goat anti-rabbit secondary antibody (1/2000; Jackson Laboratories) at RT for 1 hour. Nuclei were stained with DAPI for 10 min (1/20000; Invitrogen). Slides were mounted with Vectashield mounting medium (Vector Laboratories). All images were captured with an Axio Scan.Z1 (Zeiss) automated slide scanner or an Axiocam-Fluo microscope (Zeiss). Quantification was performed using the Qpath v0.4.3 software. The percentage of positive cells was determined by dividing the positively stained surface by total lesion area, or the number of LYVE[+] lymphatic vessels or the number of CD3[+] cells in atherosclerotic lesions were counted.

## Biochemical analysis of serum

A Cobas C111 analyzer (Roche Diagnostics) was used to measure total cholesterol, LDL, HDL, triglyceride (TG) and free fatty acid (FFA) levels in mouse sera before and after HCD.

## Quantitative PCR

LEC extraction and culture was performed using previously described protocols [18, 21]. Briefly, dermal LNs were collected, pooled and incubated in RPMI-1640 medium (Gibco) containing 0.8 mg/mL dispase (Gibco), 0.2 mg/mL collagenase-P (Roche) and 0.1 mg/mL DNase-1 (Roche) for digestion. After centrifugation, extracted cells were filtered through a 70 μm cell strainer and 5 x $10^6$ cells were seeded in 6-well plates coated with 10 μg/mL human fibronectin (Merck) and 10 μg/mL Purecol (Cellsystems). Culture medium (alpha-MEM; Gibco) was renewed every other day. After 7 days cells were detached by incubating them with Accutase (5 min; 37˚C; Gibco), LECs were sorted by flow cytometry using previously established protocols [18] and total RNA was extracted using a RNAqueous-Micro Kit (Thermo Fisher) according to the manufacturer's instructions. cDNA was obtained using the Quantitect Reverse Transcription kit (Qiagen) and real-time PCR was performed with a QuantStudio™ 6 Pro (Thermo Fisher) using the TaqMan Fast Universal master mix (Applied Biosystem). Mouse vascular cell adhesion molecule 1 (VCAM-1) or 18S primers and probes were obtained from Applied Biosystems.

## Statistical analysis

Statistical analyses were performed with Graphpad Prism 9 software and results were reported as mean ± SEM. Two-group comparisons were performed using the Student's *t*-test. Multiple groups comparisons were performed using ANOVA with Bonferroni's post-test. Differences with $P \leq 0.05$ were considered as statistically significant.

## Results

### Panx1 deficiency in LECs does not affect the number of adventitial lymphatic vessels

To investigate a potential contribution of Panx1 in LECs in the initiation of atherosclerosis, we induced Panx1 deletion with tamoxifen in 6 weeks-old *Panx1*<sup>LECdel</sup>*Apoe*<sup>-/-</sup> mice and control *Panx1*<sup>fl/fl</sup>*Apoe*<sup>-/-</sup> mice. Four weeks later, when mice reached 10 weeks of age, we induced atherosclerosis by feeding the mice with a HCD for 6 weeks (Fig 1A). To determine whether Panx1 deletion from LECs would affect the number of lymphatic vessels in atherosclerotic lesions, we performed LYVE-1 immunofluorescent staining on aortic roots of both male and female *Panx1*<sup>LECdel</sup>*Apoe*<sup>-/-</sup> and control *Panx1*<sup>fl/fl</sup>*Apoe*<sup>-/-</sup> mice. Lymphatic vessels (stained in red) were typically found in the adventitial layer of the aorta, beneath the atherosclerotic lesions (Fig 2A). Neither Panx1 deletion in LECs nor the sex of the mice did affect the number of lymphatic vessels at this location (Fig 2B). These results confirm the presence of lymphatic vessels in the adventitia of the aorta at the level of the aortic valves, our principal study location. Moreover, they demonstrate that Panx1 deletion or that the sex of the mice did not affect the amount of lymphatic vessels present during the early phases of atherosclerotic plaque development.

### Male mice have higher levels of serum triglycerides and free fatty acids

Body weight and serum lipid levels were measured before and after HCD (Fig 1A). As expected, male mice were heavier than female mice irrespective of their genotype. All mice

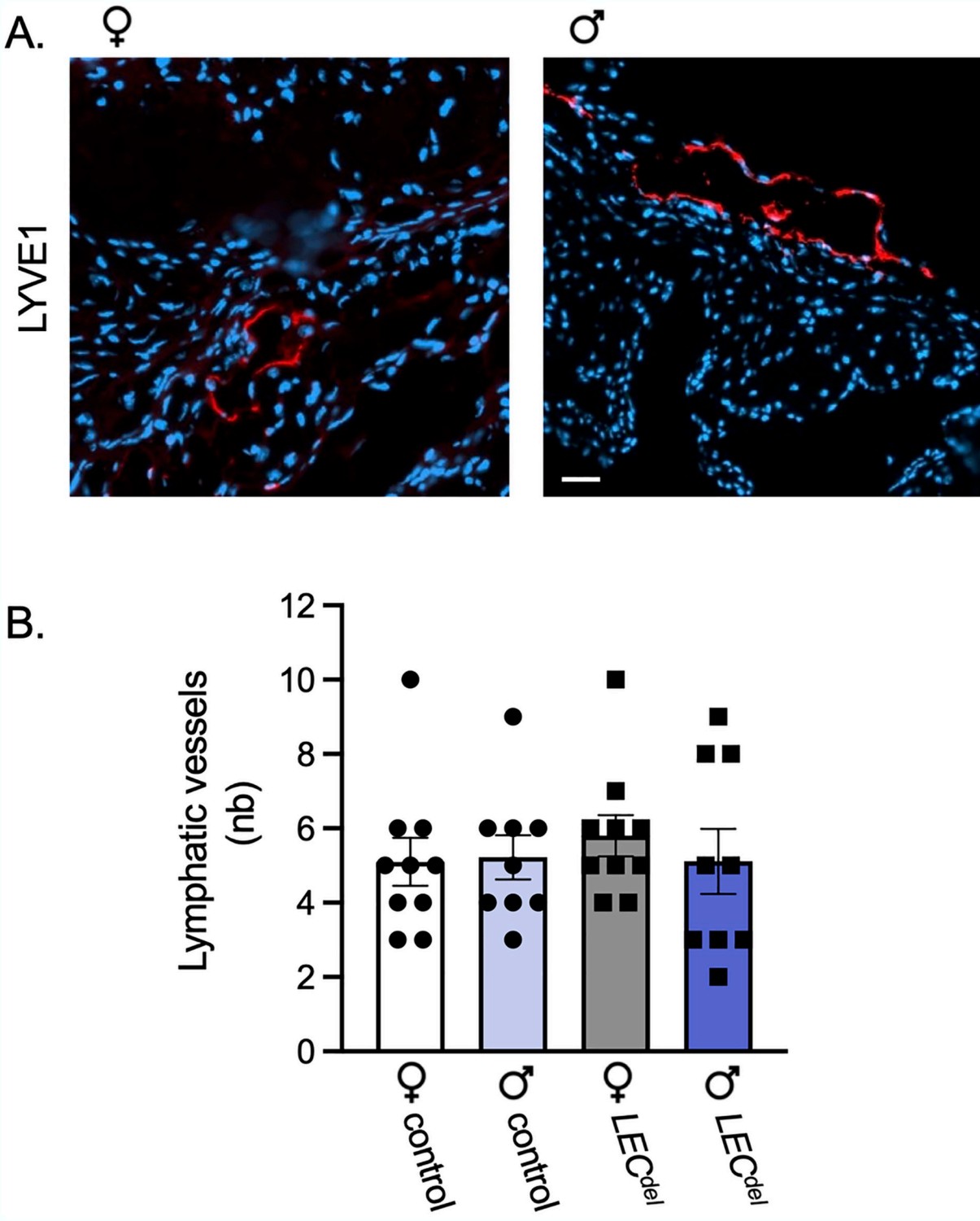

**Fig 2. Adventitial lymphatic vessel number is similar in male and female *Panx1^LECdel^Apoe^-/-* and control mice.** (A) Representative images of LYVE1 immunofluorescent staining (red) in aortic roots. Nuclei were stained with DAPI (blue). Scale bar represents 50 μm. (B) The lymphatic vessel number in the aortic roots was counted in female control *Panx1^fl/fl^Apoe^-/-* mice (white), female *Panx1^LECdel^Apoe^-/-* mice (grey), male control *Panx1^fl/fl^Apoe^-/-* mice (light blue) and male *Panx1^LECdel^Apoe^-/-* mice (dark blue). Mean ± SEM, N = 9–10.

significantly increased their weight during HCD, but no differences in weight gain were observed between males and females nor between $Panx1^{fl/fl}Apoe^{-/-}$ and $Panx1^{LECdel}Apoe^{-/-}$ mice (Fig 3A). In addition, all mice showed an increase in total serum cholesterol, LDL and HDL, irrespective of their genotype or sex (Fig 3B–3D). However, we observed higher levels of serum TG and FFA before and after HCD in male $Panx1^{LECdel}Apoe^{-/-}$ and control $Panx1^{fl/fl}Apoe^{-/-}$ mice compared to females of the same genotypes (Fig 3E and 3F). These results point to sex-dependent differences in lipid metabolism between male and female mice under HCD during the initiation of atherosclerosis.

## Female mice develop larger atherosclerotic plaques

Next, we assessed whether Panx1 deletion in LECs affects the atherosclerotic burden in female and male $Panx1^{LECdel}Apoe^{-/-}$ and $Panx1^{fl/fl}Apoe^{-/-}$ mice after 6 weeks of HCD. To this end, Sudan-IV staining was performed on cryosections of aortic roots and on longitudinally opened thoraco-abdominal aortas of all mice (Fig 4A). In both $Panx1^{LECdel}Apoe^{-/-}$ and control $Panx1^{fl/fl}Apoe^{-/-}$ mice, we found larger atherosclerotic lesions in aortic roots from female mice compared to male mice of the same genotype (Fig 4B). However, the atherosclerotic burden was not different in thoracic-abdominal aortas from male and female mice of both genotypes (Fig 4C). As atherosclerotic lesions develop earlier at the level of the aortic roots [22], potential differences in atherosclerotic burden can be quicker detected at this location. These observations point to a promoting effect of female sex on early atherosclerotic plaque burden and this seems independent of serum lipid levels.

## Atherosclerotic plaque vulnerability is increased in male mice

Atherosclerotic plaque rupture can occur due to a decline in its mechanical stability. Histopathological analysis of atherosclerotic lesions associated with fatal myocardial infarction revealed a substantial lipid core, a thin fibrous cap, an accumulation of activated macrophages producing matrix metalloproteinases and a reduction in SMCs responsible for synthesizing extracellular matrix [23]. We evaluated whether Panx1 in LECs affects the phenotype of atherosclerotic lesions by analyzing the plaque composition in $Panx1^{LECdel}Apoe^{-/-}$ and control $Panx1^{fl/fl}Apoe^{-/-}$ mice after 6 weeks of HCD. No differences were found in the amount of the plaque stabilizing factors, collagen (in blue; Fig 5A and 5B) and SMCs (in green; Fig 5C and 5D), between mice of both sexes and genotypes. Interestingly, we observed differences in the amount of the plaque destabilizing factors, lipids (in red; Fig 5E and 5F) and CD68[+] macrophages (in brown; Fig 5G and 5H), i.e. the area covered by lipids and inflammatory cells was increased in male mice compared to female mice and this was independent of the genotype. Thus, male mice have more vulnerable plaques characterized by an increased content of destabilizing factors.

In summary, LEC-specific deletion of Panx1 did not alter body weight, serum lipid levels, atherosclerotic plaque burden and composition during early atherosclerosis. However, male mice had increased serum levels of TG and FFA, with or without HCD, which may have contributed to an increased lipid content in their atherosclerotic plaques attracting more macrophages and leading to a more vulnerable lesion phenotype. On the other hand, the early atherosclerotic plaque burden was higher in female mice.

## Role of Panx1 in LECs in the progression of atherosclerosis

To investigate a potential contribution of Panx1 in LECs in the progression of atherosclerosis, we placed the mice on a HCD for 4 weeks to initiate the disease, then deleted Panx1 channels in LECs and studied the advanced atherosclerotic plaque burden in the aortic roots after 6

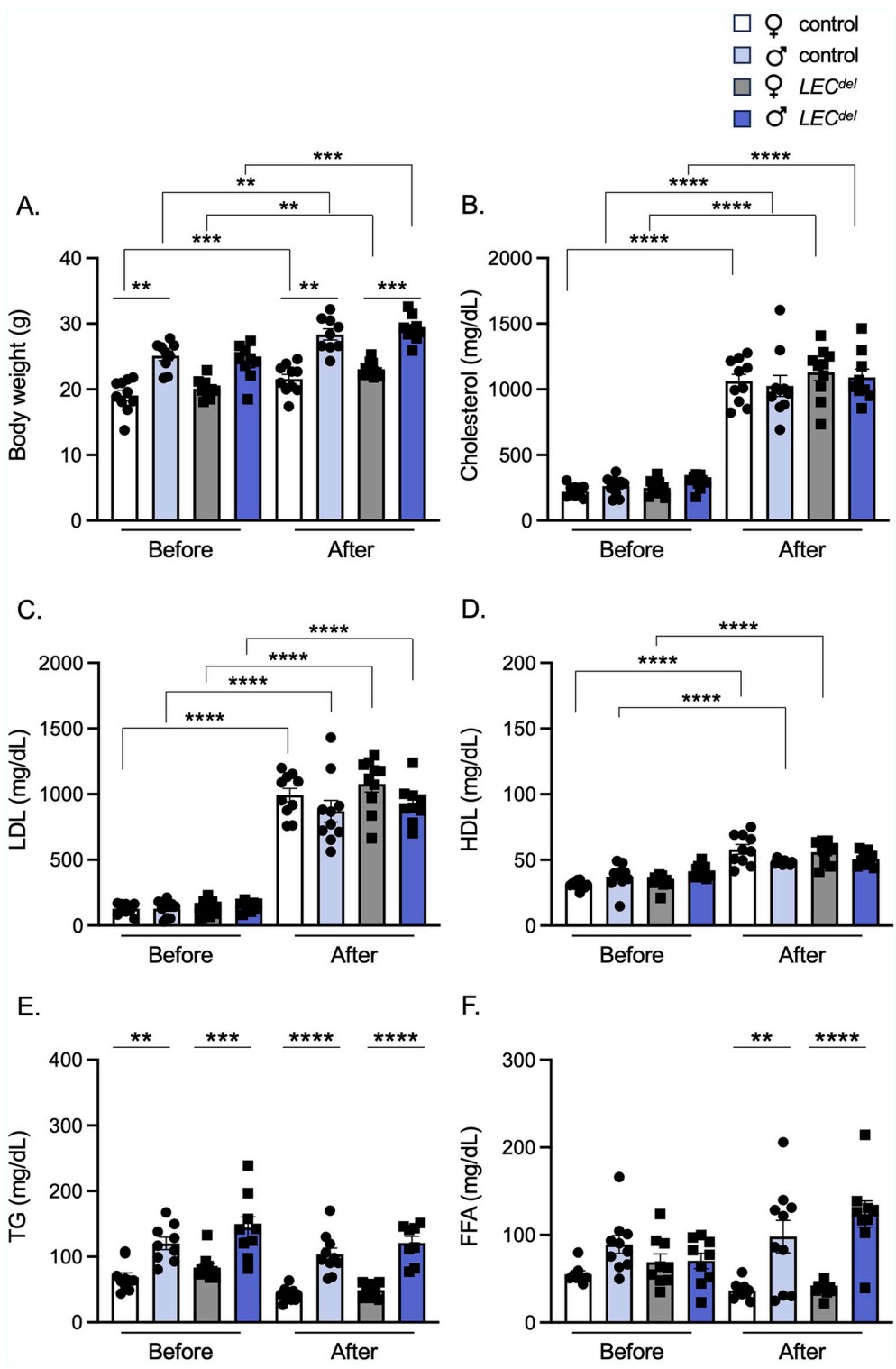

**Fig 3. Males have higher serum levels of TG and FFA after 6 weeks of HCD.** Body weight (A), serum cholesterol (B), LDL (C), HDL (D), TG (E) and FFA (F) levels of female control $Panx1^{fl/fl}Apoe^{-/-}$ mice (white), female $Panx1^{LECdel}Apoe^{-/-}$ mice (grey), male control $Panx1^{fl/fl}Apoe^{-/-}$ mice (light blue) and male $Panx1^{LECdel}Apoe^{-/-}$ mice (dark blue) before and after 6 weeks of HCD. Mean ± SEM, N = 8–10, **$P\leq0.01$, ***$P\leq0.001$, ****$P\leq0.0001$.

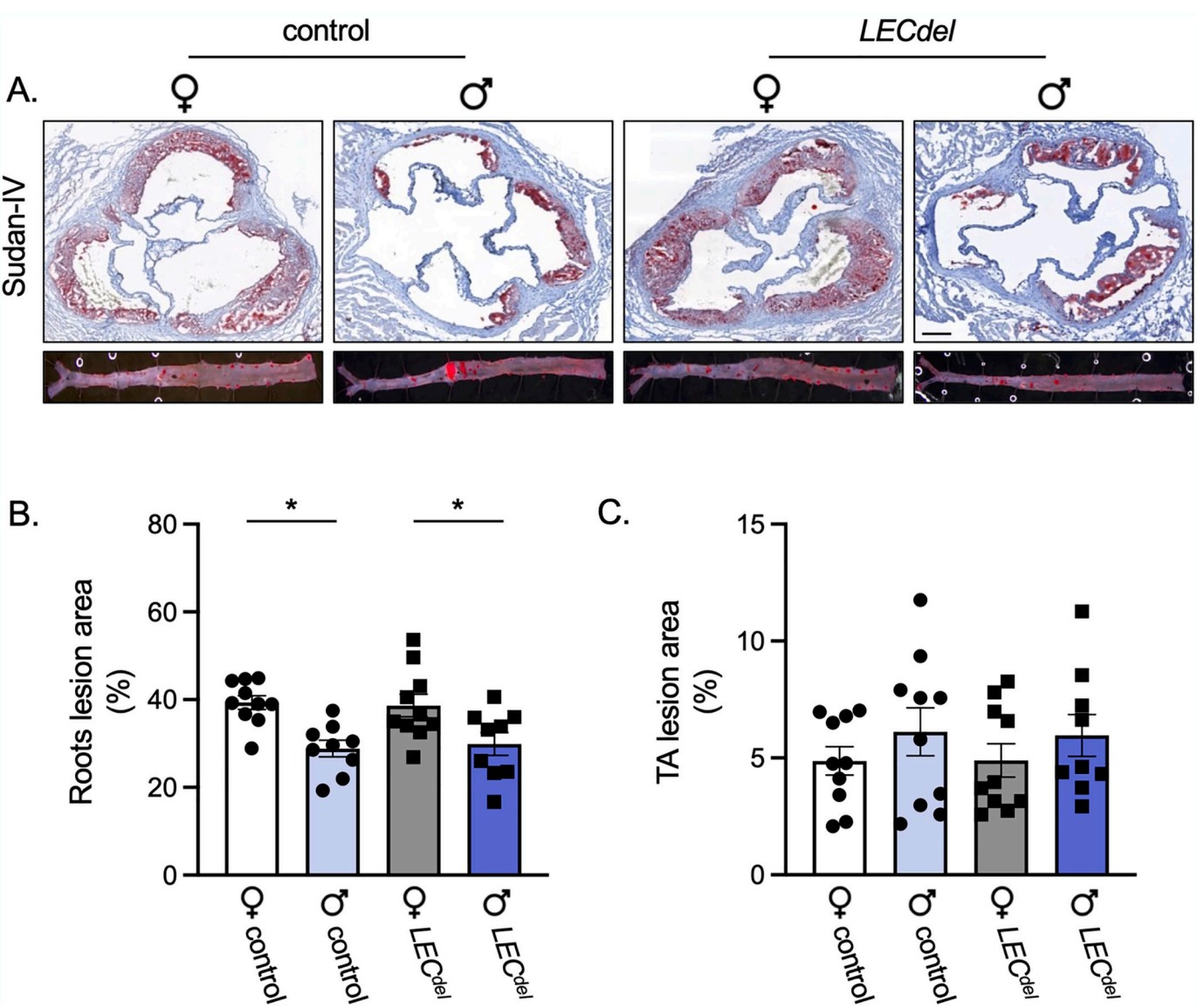

**Fig 4. Females have larger atherosclerotic plaques in aortic roots after 6 weeks of HCD.** Sudan-IV staining (A) and quantification of atherosclerotic plaque burden in aortic roots (B) and thoraco-abdominal aortas (C) of female control $Panx1^{fl/fl}Apoe^{-/-}$ mice (white), female $Panx1^{LECdel}Apoe^{-/-}$ mice (grey), male control $Panx1^{fl/fl}Apoe^{-/-}$ mice (light blue) and male $Panx1^{LECdel}Apoe^{-/-}$ mice (dark blue) after 6 weeks of HCD. Scale bar represents 200 µm. Mean ± SEM, N = 9–10, *$P \leq 0.05$.

more weeks of HCD (Fig 1B). As expected, the atherosclerosis burden in the aortic roots was increased for all 4 groups after 10 weeks of HCD (Fig 6A and 6B) compared with 6 weeks of HCD (Fig 4C). Moreover, the extent of atherosclerosis after 10 weeks of HCD was higher in female $Panx1^{LECdel}Apoe^{-/-}$ mice than in males of the same genotype. This difference was not observed in $Panx1^{fl/fl}Apoe^{-/-}$ mice and seemed thus to result rather from LEC-specific Panx1 deletion than from sex-dependent differences in advanced atherosclerosis (Fig 6B). Thus, while the early atherosclerotic plaque burden was higher in female mice, this difference was no longer present when plaques advanced. Finally, as the effects of Panx1 deletion on lymphatic function are known to be more subtle than the sizeable effects of sex on cardiovascular disease,

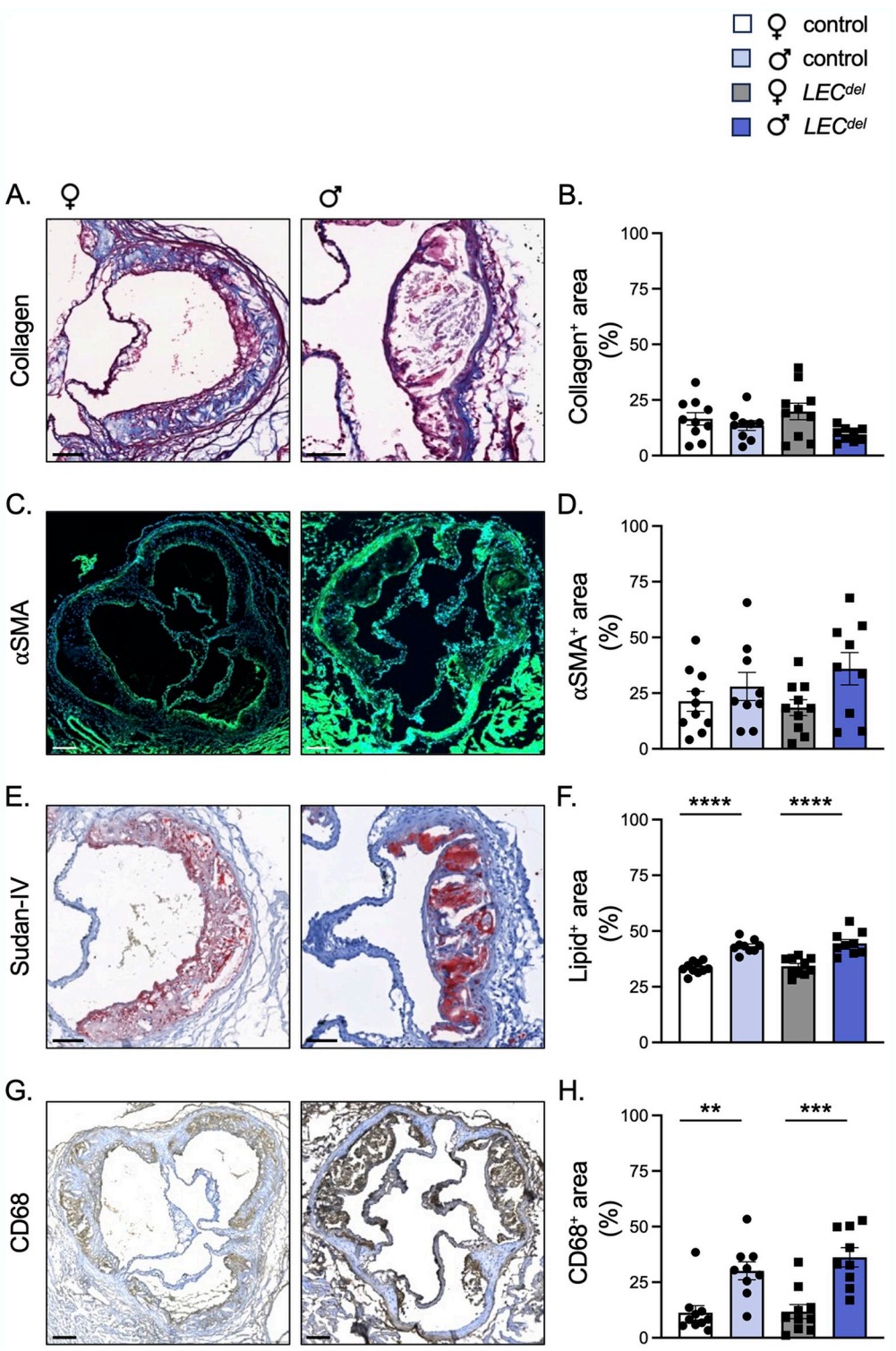

**Fig 5. Reduced atherosclerotic plaque stability in male mice after 6 weeks of HCD.** Representative images and quantification of (immuno-) stainings for collagen (A,B; in blue), α-SMA (C,D; in green), Sudan-IV (E,F; in red) and CD68 (G,H; in brown) performed on aortic roots of female control *Panx1^{fl/fl}Apoe^{-/-}* mice (white), female *Panx1^{LECdel}Apoe^{-/-}* mice (grey), male control *Panx1^{fl/fl}Apoe^{-/-}* mice (light blue) and male *Panx1^{LECdel}Apoe^{-/-}* mice (dark blue) after 6 weeks of HCD. Scale bars represent 100 μm for A and E, and 200 μm for C and G. Mean ± SEM, N = 9–10, **$P \leq 0.01$, ***$P \leq 0.001$, ****$P \leq 0.0001$.

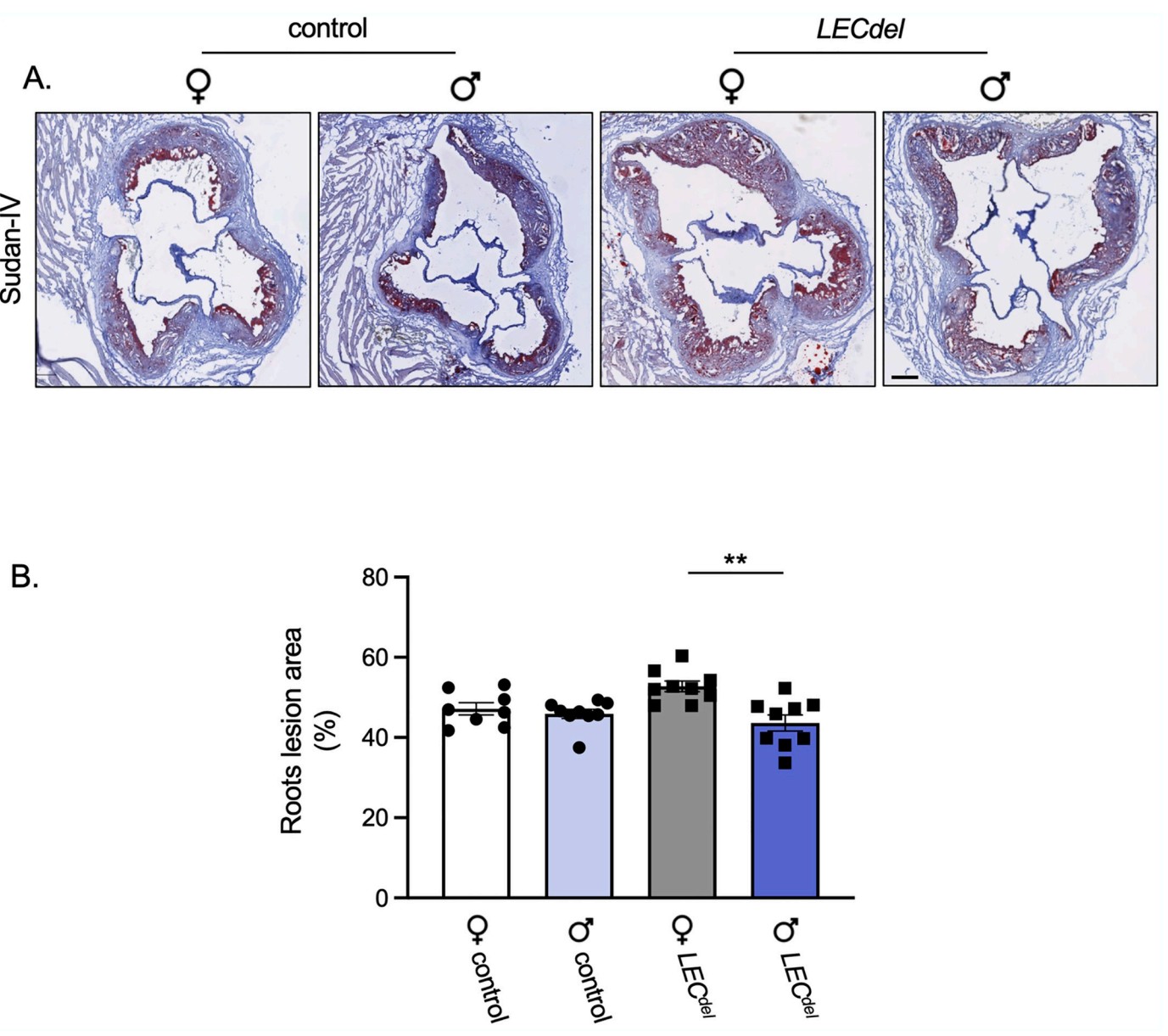

**Fig 6. Increased advanced atherosclerotic plaque burden in female *Panx1^{LECdel}Apoe^{-/-}* mice.** Sudan-IV staining (A) and quantification of atherosclerotic lesion extent in the aortic roots (B) of female control *Panx1^{fl/fl}Apoe^{-/-}* mice (white), female *Panx1^{LECdel}Apoe^{-/-}* mice (grey), male control *Panx1^{fl/fl}Apoe^{-/-}* mice (light blue) and male *Panx1^{LECdel}Apoe^{-/-}* mice (dark blue) after 10 weeks of HCD. Scale bar represents 200 μm. Mean ± SEM, N = 8–9, **$P \leq 0.01$.

we analyzed results onto the effects of LEC-specific Panx1 deletion on the progression of atherosclerosis separately for male and female mice.

The body weight (Fig 7A), serum cholesterol (Fig 7B) and LDL (Fig 7C) levels were similarly increased in male *Panx1^{LECdel}Apoe^{-/-}* and *Panx1^{fl/fl}Apoe^{-/-}* mice after 10 weeks of HCD. Likewise, we did not observe any difference in serum HDL (Fig 7D), TG (Fig 7E) and FFA (Fig 7F) levels between male *Panx1^{LECdel}Apoe^{-/-}* and control *Panx1^{fl/fl}Apoe^{-/-}* mice after the 10 weeks of HCD. This implies that inducing Panx1 deletion in LECs did not affect circulating lipid concentrations during the progression of atherosclerosis.

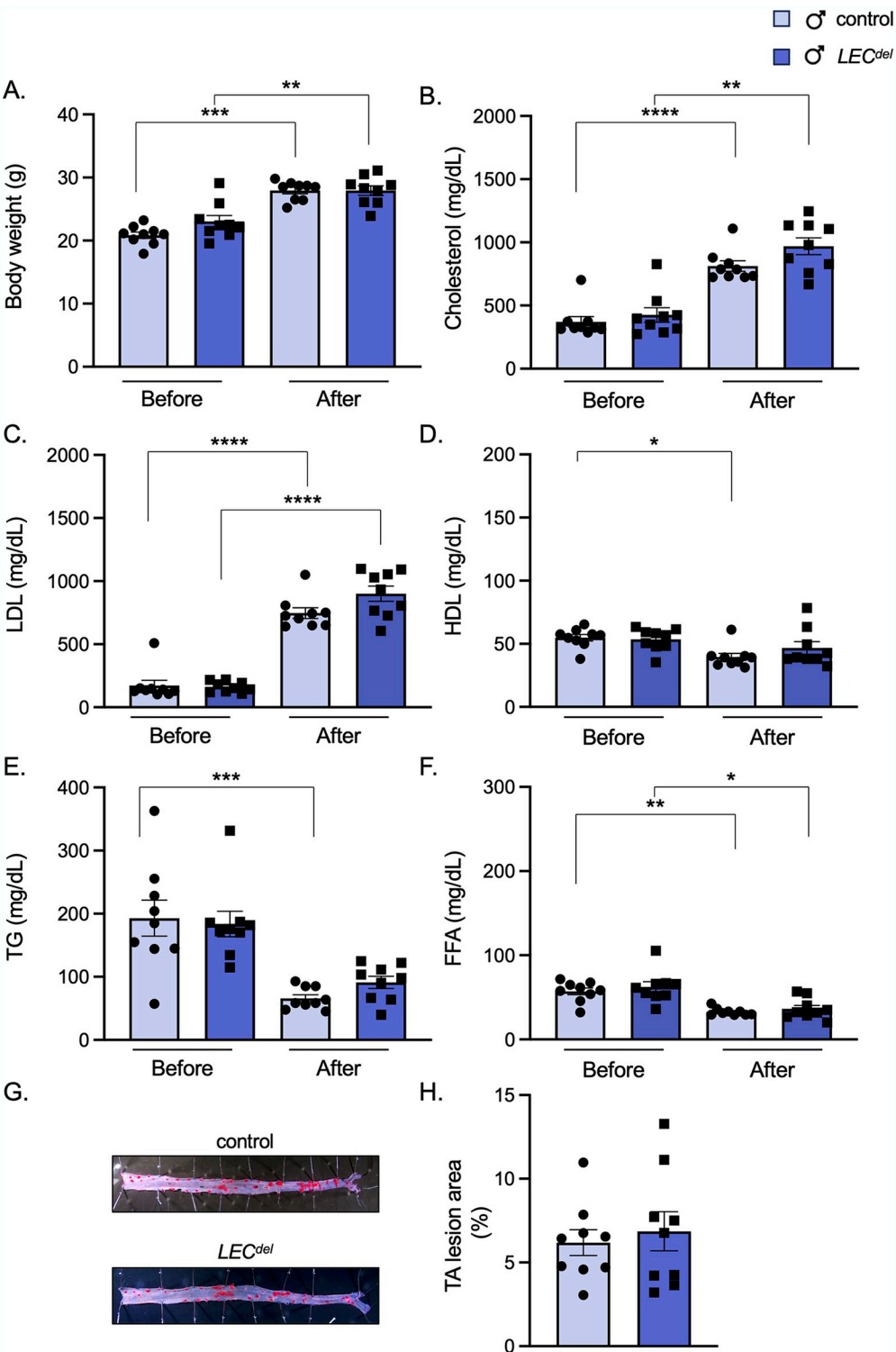

**Fig 7. Panx1 deletion in LECs does not affect serum lipid levels and advanced atherosclerotic plaque burden in males.** Body weight (A), serum cholesterol (B), LDL (C), HDL (D), TG (E) and FFA (F) levels of male control $Panx1^{fl/fl}Apoe^{-/-}$ mice (light blue) and male $Panx1^{LECdel}Apoe^{-/-}$ mice (dark blue) after 10 weeks of HCD. Sudan-IV staining (E) and quantification of atherosclerotic lesion burden in thoraco-abdominal aortas (F) of male $Panx1^{fl/fl}Apoe^{-/-}$ mice (light blue) and male $Panx1^{LECdel}Apoe^{-/-}$ mice (dark blue) after 10 weeks of HCD. Mean ± SEM, N = 9, *$P \leq 0.05$, **$P \leq 0.01$, ***$P \leq 0.001$, ****$P \leq 0.0001$.

Next, we analyzed the atherosclerotic plaque burden in the thoracic-abdominal aortas in male *Panx1^{fl/fl}Apoe^{-/-}* and *Panx1^{LECdel}Apoe^{-/-}* mice after 10 weeks of HCD. Similar to the observations during early atherosclerosis (Fig 4), we found no effect of LEC-specific Panx1 deletion on lesion burden in the thoracic-abdominal aortas (Fig 7G and 7H) nor in the aortic roots (Fig 6B) of the mice after 10 weeks of HCD. This suggests that Panx1 in LECs does not play a substantial role in modulating the initiation and the progression of atherosclerotic lesions in male mice.

Finally, we investigated whether Panx1 deletion in LECs during plaque development may affect the composition of advanced atherosclerotic plaques in male *Panx1^{LECdel}Apoe^{-/-}* mice. Panx1 deletion had no impact on the phenotype of the lesions in male mice, neither for the stabilizing factors such as collagen (Fig 8A) and SMCs (Fig 8B) nor for the destabilizing factors like lipids (Fig 8C) and CD68^+ macrophages (Fig 8D). The number of T lymphocytes in the advanced atherosclerotic lesions was also not different between *Panx1^{LECdel}Apoe^{-/-}* and control *Panx1^{fl/fl}Apoe^{-/-}* mice (Fig 8E). Despite the sex-specific favorable early plaque phenotype in male mice, the deletion of Panx1 from LECs during atherosclerotic plaque progression had no effect on the stability of atherosclerotic lesions in male mice.

To investigate if deletion of Panx1 channels in LECs affects the progression of atherosclerosis in female mice, we used the same protocol as for the male mice (Fig 1B). Female *Panx1^{LECdel}Apoe^{-/-}* and control *Panx1^{fl/fl}Apoe^{-/-}* mice had a similar body weight before the HCD and their weight gain after the 10 weeks of HCD was not different (Fig 9A). Likewise, the serum cholesterol levels of female *Panx1^{LECdel}Apoe^{-/-}* and *Panx1^{fl/fl}Apoe^{-/-}* mice were not different before the HCD and they showed a similar increase following the 10 weeks of HCD (Fig 9B). Along the same line, serum LDL (Fig 9C), HDL (Fig 9D), TG (Fig 9E) and FFA (Fig 9F) levels did not differ between the two genotypes before and after HCD. Similar to the results obtained in male mice, inducing Panx1 deletion in LECs did not affect the serum lipids in female mice during the progression of atherosclerosis.

The extent of the atherosclerotic lesions tended to be increased in female *Panx1^{LECdel}Apoe^{-/-}* mice compared to *Panx1^{fl/fl}Apoe^{-/-}* control mice at the level of the aortic roots (Fig 6B) and we observed a similar trend towards increased plaque burden in the thoracic-abdominal aortas of female *Panx1^{LECdel}Apoe^{-/-}* mice (Fig 9G and 9H). These results suggest an atheroprotective role for Panx1 in LECs in female mice during the progression of atherosclerosis.

Finally, we examined whether Panx1 deletion in LECs during plaque development affects the composition of advanced atherosclerotic plaques in female *Panx1^{LECdel}Apoe^{-/-}* and *Panx1^{fl/fl}Apoe^{-/-}* mice. We did not observe genotype-dependent differences in female mice with respect to the plaque stabilizing collagen and SMC content (Fig 10A and 10B) and the plaque destabilizing lipid and macrophage content (Fig 10C and 10D) in advanced atherosclerotic plaques. However, the number of T lymphocytes in the advanced atherosclerotic lesions was increased in female *Panx1^{LECdel}Apoe^{-/-}* mice as compared to control *Panx1^{fl/fl}Apoe^{-/-}* mice of the same sex (Fig 10E). Interestingly, the expression level of VCAM-1 was lower in Panx1-deficient LECs than in Panx1-expressing LECs isolated from LNs of female mice (Fig 10F). In summary, female *Panx1^{LECdel}Apoe^{-/-}* mice tend to have more atherosclerotic lesions that contained more T lymphocytes than the plaques of control *Panx1^{fl/fl}Apoe^{-/-}* mice.

## Discussion

Panx1 expression is higher in LECs of male mice than in the ones of female mice [18]. Panx1 deletion in LECs is known to alter important lymphatic functions such as tissue fluid drainage, dietary fat uptake and DC migration to draining LNs in a sex-specific manner [18]. The

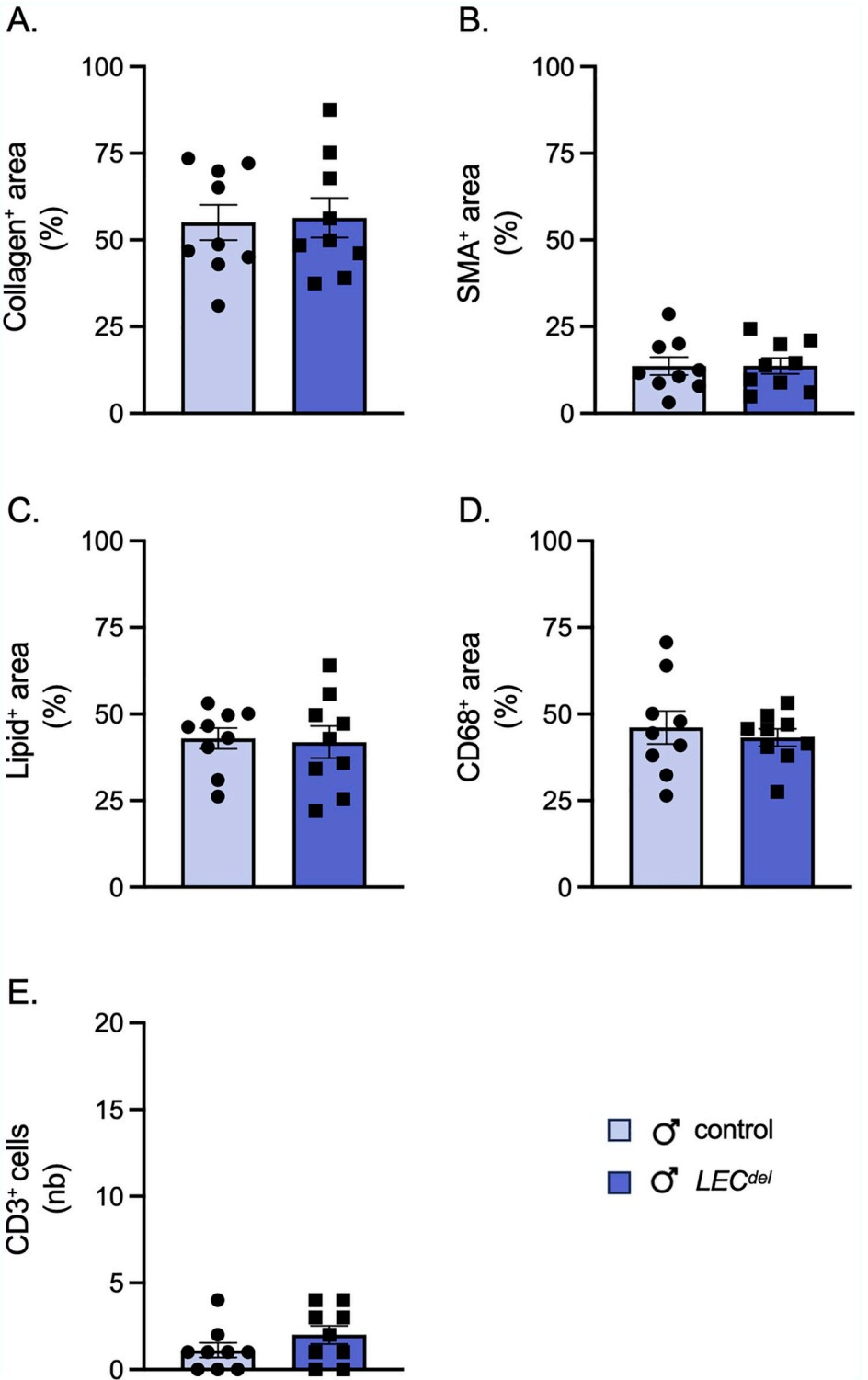

**Fig 8. Panx1 deletion in LECs does not affect advanced plaque stability in males.** Quantification of (immuno-) stainings for collagen (A), α-SMA (B), Sudan-IV (C), CD68 (D) and CD3 (E) in aortic roots of male $Panx1^{fl/fl}Apoe^{-/-}$ mice (light blue) and male $Panx1^{LECdel}Apoe^{-/-}$ mice (dark blue) after 10 weeks of HCD. Mean ± SEM, N = 9.

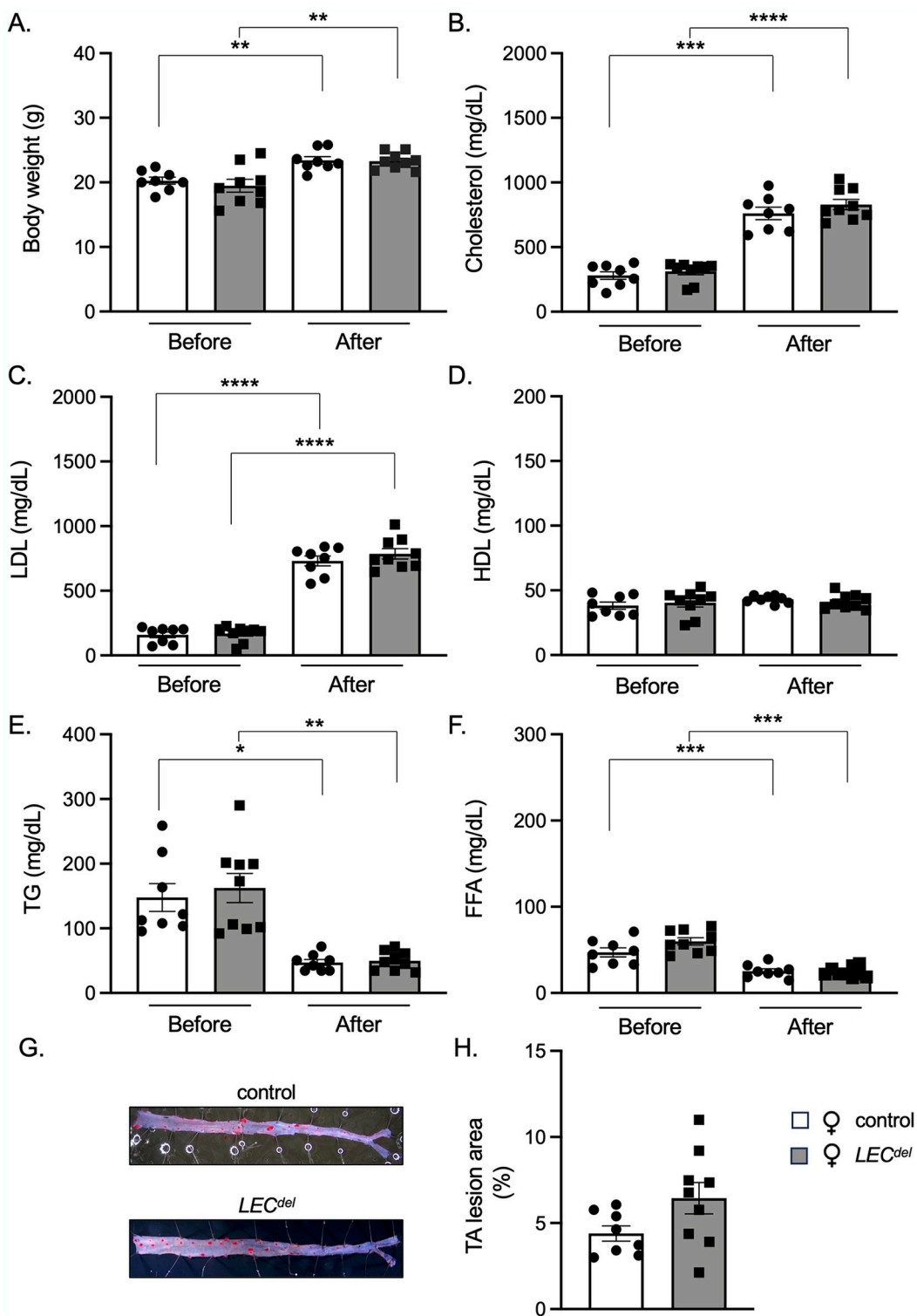

**Fig 9. Panx1 deletion in LECs enhances advanced atherosclerosis in aortic roots of female mice.** Body weight (A), serum cholesterol (B), LDL (C), HDL (D), TG (E) and FFA (F) levels of female control *Panx1^{fl/fl}Apoe^{-/-}* mice (white) and female *Panx1^{LECdel}Apoe^{-/-}* mice (grey) after 10 weeks of HCD. Sudan-IV staining (E) and quantification of atherosclerotic lesion burden in thoraco-abdominal aortas (F) of male *Panx1^{fl/fl}Apoe^{-/-}* mice (white) and male *Panx1^{LECdel}Apoe^{-/-}* mice (grey) after 10 weeks of HCD. Mean ± SEM, N = 8–9, *$P \leq 0.05$, **$P \leq 0.01$, ***$P \leq 0.001$, ****$P \leq 0.0001$.

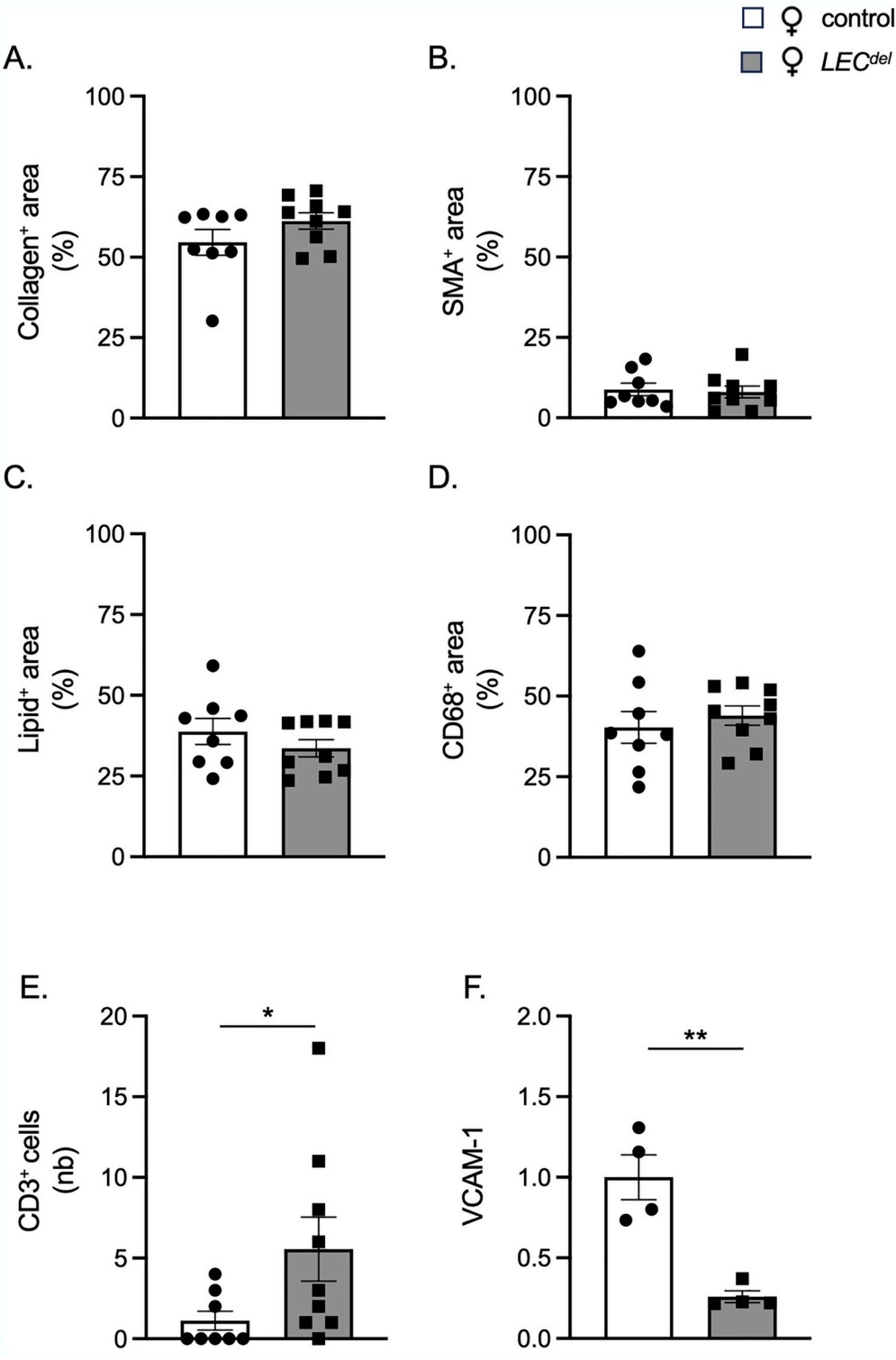

**Fig 10. Panx1 deletion in LECs results in T cell enrichment in advanced atherosclerotic lesions of female mice.**
Quantification of (immuno-)stainings for collagen (A), α-SMA (B), Sudan-IV (C), CD68 (D) and CD3 (E) in aortic roots of female *Panx1^fl/fl^Apoe^-/-^* mice (white) and female *Panx1^LECdel^Apoe^-/-^* mice (grey) after 10 weeks of HCD. Mean ± SEM, N = 8–9. VCAM-1 expression (F) in Panx1-expressing (white) and Panx1-deficient (grey) LECs isolated from LNs of female mice. Mean ± SEM, n = 4. *$P \leq 0.05$, **$P \leq 0.01$.

present study was undertaken to study the effects of Panx1 deletion in LECs on the initiation and the progression of atherosclerosis in male and female mice.

Lymphatic vessels are typically found in the adventitial layer beneath atherosclerotic plaques and close to the *vasa vasorum* [24], which was also the case in our study (Fig 2). Panx1 deletion in LECs did not affect the number of adventitial lymphatic vessels both in male and in female mice. Although an earlier study showed that Panx1 regulated lymphangiogenesis *in vitro* [14], our results did not confirm these results *in vivo*, suggesting a minor or no role of LEC Panx1 in lymphangiogenesis in atherosclerotic lesions. The relation between Panx1 in LECs and atherosclerosis may however go beyond quantitative changes in lymphatic vessel number and further experiments focused on the implications of LEC-specific Panx1 deletion on atherosclerotic plaque burden and stability during the initiation phase as well as during the progression of the disease.

During early atherosclerosis induced by 6 weeks of HCD, we observed mainly sex-dependent differences rather than differences induced by Panx1 gene deletion. As expected, the body weight was higher in males than in females (Fig 3A) [25]. Both males and females increased their serum cholesterol, LDL and HDL levels to a similar extent in response to the HCD (Fig 3B and 3D). However, serum TG and FFA levels were higher in male *Panx1$^{fl/fl}$Apoe$^{-/-}$* and *Panx1$^{LECdel}$Apoe$^{-/-}$* mice compared to females (Fig 3E and 3F), which points to a potential role of sex hormones in lipid metabolism. This may be linked to sex-dependent differences in clearance of TG and fat storage. It has been shown that estradiol promotes the accumulation of subcutaneous fat and that estrogen deficiency in women leads to weight gain [26]. Moreover, another study revealed that women produce more TG-rich very low-density lipoproteins (VLDL), which are matched by higher rates of lipoprotein lipase-mediated VLDL clearance, causing an overall lower blood TG level in women [27]. The difference in dietary lipid processing between male and female mice may also result from sex-dependent changes in the trafficking of dietary fats through lymphatic vessels. Using a well-established lymphatic fistula model, it has been shown that female mice displayed a lower lymphatic flow of triacylglycerol than male mice [28]. Further to the differences in serum TG and FFA levels, we found that female mice had an increased early atherosclerotic burden in the aortic roots compared to males (Fig 4B), but the male plaques were more vulnerable due to increased content of macrophages and lipids (Fig 5E–5H). This is in accordance with a previous study in which male *Apoe$^{-/-}$* mice showed smaller atherosclerotic plaques in the aortic roots, but expressed higher levels of pro-inflammatory macrophage markers than females [29]. Moreover, it illustrates that the size of a lesion does not necessarily correlate with its vulnerability (*i.e.* probability to rupture), corroborating an observation that is long known from clinical studies [30]. It also highlights that in mouse models of atherosclerosis, the plaque inflammatory status rather than the plaque size may be closer to sex-specific clinical outcomes found in humans due to its contribution to plaque vulnerability [29].

Limiting plaque progression, or even inducing plaque regression, may involve a reduction in plaque lipid content, macrophage and T cell content and inflammatory status [31], which are all processes in which the lymphatic system has a regulatory function. In an attempt to mimic as closely as possible a potential clinical treatment situation involving Panx1 channels, we used a study protocol in which atherosclerosis was initiated by 4 weeks of HCD, and Panx1 channels were subsequently deleted from the lymphatic endothelium while the HCD continued for another 6 weeks. As the male early atherosclerotic plaques appeared more vulnerable than female plaques (Fig 5), we first examined the impact of Panx1 deletion in LECs in male mice. We observed no differences between the male *Panx1$^{LECdel}$Apoe$^{-/-}$* and control *Panx1$^{fl/fl}$Apoe$^{-/-}$* mice in terms of serum lipid levels (Fig 7), atherosclerotic plaque burden after 10 weeks of HCD (Figs 6 and 7) or advanced plaque stability (Fig 8). These results led

us to consider the possibility that a compensatory mechanism may be in place to overcome the effects of Panx1 deficiency in LECs during the progression of the pathology. We checked for the expression of the other members of the pannexin family, Panx2 and Panx3, and did not detect these mRNAs in LECs (data not shown). A hypothetical compensatory mechanism could also involve connexins (Cxs), a family of channel-forming proteins involved in direct cell-to-cell communication [32]. Cxs form hexamers, called connexons, and gap junction channels are formed by the docking of connexons from two adjacent cells. Similar to Panx1 channels, however, connexons can also function as hemichannels allowing the diffusion of small molecules and ions between the cytoplasm and the extracellular environment [33]. Among the Cx family members, Cx47 has been associated to lymphedema [34–36] and it is known to be expressed in LECs [37–39]. Moreover, Cx47 was shown to regulate serum lipid levels, thereby not altering the atherosclerotic burden but rather plaque composition [39]. Further experiments might elucidate a possible compensatory role of Cx47 in the context of Panx1 deletion and atherosclerosis. Such experiments are however beyond the scope of this study.

In contrast to male mice, female $Panx1^{LECdel}Apoe^{-/-}$ mice showed a tendency to a larger atherosclerotic plaque burden after 10 weeks of HCD than control $Panx1^{fl/fl}Apoe^{-/-}$ mice (Figs 6 and 9G and 9H), despite similar serum lipid levels in both genotypes (Fig 9B–9F). Adventitial lymphatic vessels beneath advanced atherosclerotic lesions drain interstitial fluid and immune cells, particularly T cells and dendritic cells, from the vessel wall to draining lymph nodes [13]. Dysfunction or insufficient number of adventitial lymphatic vessels can exacerbate atherosclerotic plaque progression by preventing proper immune cell clearance and fostering persistent inflammation. Although the number of lymphatic vessels beneath advanced atherosclerotic lesions was similar for both genotypes and sexes (Fig 2), atherosclerotic plaques of female $Panx1^{LECdel}Apoe^{-/-}$ mice, but not of male $Panx1^{LECdel}Apoe^{-/-}$ mice, contained more T lymphocytes than control $Panx1^{fl/fl}Apoe^{-/-}$ mice (Figs 8E and 10E). This suggests impaired T cells drainage from female atherosclerotic plaques by lymphatic vessels lacking Panx1 expression in LECs. In agreement, we found a reduced expression of VCAM-1 in female Panx1-deficient LECs compared to female Panx1-expressing LECs (Fig 10F), further supporting the hypothesis of a sex-specific regulation of T cell drainage by Panx1 expression in LECs. However, whether the sex-specific decrease in T cell drainage results in the increased atherosclerotic plaque burden in female $Panx1^{LECdel}$ mice still remains to be proven.

In conclusion, while Panx1 deletion did not influence the early stages of atherosclerosis, it emerged as a factor in modulating disease progression in females. Independent of Panx1 expression, a sex-specific difference in early atherogenesis was also found in this study with a larger atherosclerotic plaque burden in female compared to the male mice. Together these observations suggest that lymphatic Panx1 exerts a protective effect against atherosclerosis progression in female mice, highlighting the complex interplay between sex-specific factors and disease pathogenesis.

## Acknowledgments

We thank Viviane Bes for excellent technical assistance and advice.

## Author Contributions

**Conceptualization:** Filippo Molica, Brenda R. Kwak.

**Data curation:** Filippo Molica, Brenda R. Kwak.

**Formal analysis:** Avigail Ehrlich, Filippo Molica.

**Funding acquisition:** Brenda R. Kwak.

**Investigation:** Avigail Ehrlich, Graziano Pelli, Bernard Foglia, Filippo Molica.

**Methodology:** Avigail Ehrlich, Graziano Pelli, Bernard Foglia, Filippo Molica.

**Project administration:** Brenda R. Kwak.

**Resources:** Brenda R. Kwak.

**Supervision:** Filippo Molica, Brenda R. Kwak.

**Validation:** Filippo Molica, Brenda R. Kwak.

**Writing – original draft:** Avigail Ehrlich.

**Writing – review & editing:** Avigail Ehrlich, Graziano Pelli, Bernard Foglia, Filippo Molica, Brenda R. Kwak.

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
