## [Decision Letter · Decision Letter 0]

4 Oct 2024

PONE-D-24-40523Atheroprotective role of Panx1 deletion in lymphatic endothelial cells in female micePLOS ONE

Dear Dr. Kwak,

Thank you for submitting your manuscript to PLOS ONE. After careful consideration, we feel that it has merit but does not fully meet PLOS ONE’s publication criteria as it currently stands. Therefore, we invite you to submit a revised version of the manuscript that addresses the points raised during the review process.

We look forward to receiving your revised manuscript.

Kind regards,

Michael Bader

Academic Editor

PLOS ONE

Journal Requirements:

https://journals.plos.org/plosone/s/file?

 This study was supported by grants from the Swiss National Science Foundation (grant number 310030_182573 to BRK) and the Gabbiani fund (to AE).  

NO authors have competing interests. 

Reviewers' comments:

Reviewer's Responses to Questions

**Comments to the Author**

1. Is the manuscript technically sound, and do the data support the conclusions?

Reviewer #1: Partly

Reviewer #2: No

2. Has the statistical analysis been performed appropriately and rigorously? 

Reviewer #1: Yes

Reviewer #2: Yes

3. Have the authors made all data underlying the findings in their manuscript fully available?

Reviewer #1: Yes

Reviewer #2: Yes

4. Is the manuscript presented in an intelligible fashion and written in standard English?

Reviewer #1: Yes

Reviewer #2: Yes

5. Review Comments to the Author

Reviewer #1: The authors investigated the function of Panx1 in lymphatic endothelial cells concerning atherosclerotic lesions and revealed that this molecule specifically regulates lesions in female mice. They concluded that this phenomenon is due to changes in the ability of lymphatic endothelial cells to transport T cells. The results are clear, without significant contradictions, and the novelty is evident, but there appears to be a lack of experimental data regarding immune cells to support the above conclusions. Specifically, the possibility that lymphatic endothelial cells induce qualitative and quantitative changes in immune cells has been overlooked.

Comments

The authors conclude that the promotion of atherosclerosis in female Prox1-CreERT2+Panx1 fl/flApoe-/- mice affects T cell accumulation in the lesions, yet this study does not assess T cell accumulation in the lesions or the populations of T cells in the blood. Since T cells are roughly classified into proinflammatory and anti-inflammatory subsets, both quantitative and qualitative evaluations are essential. Previous reports (PMID: 36519468) have indicated that dysfunction of lymphatic endothelial cells reduces the stability of anti-inflammatory regulatory T cells systemically, increasing susceptibility to atherosclerosis, suggesting that impaired cellular transport via lymphatics may not always be the primary cause. In the author’s context, evaluating lymphatic endothelial cell functions, such as adhesion molecules, is warranted.

The authors should discuss why the observed sex differences occur, in terms of Panx1. The current draft provides extensive explanations regarding gender differences in dyslipidemia; however, as these do not lead to a definitive conclusion, they should be condensed. Instead, the discussion should focus on why there are gender differences in Panx1's contribution to atherosclerosis. It would be beneficial to examine the differences in Panx1 expression between males and females.

Lipoprotein cholesterol and triglycerides should be fractionated into at least VLDL, LDL, and HDL for evaluation. Merely assessing changes in total cholesterol is insufficient to rule out the involvement of dyslipidemia.

The two experimental strategies (initiation and progression) are not clearly linked to the figures, even in the legends. Please revise this for clarity.

Reviewer #2: The goal of this manuscript is to examine the impact of lymphatic endothelial Pannexin 1 (LEC Panx1) on the development and stability of atherosclerotic plaques. Despite being an intriguing subject, the data provided does not indicate a role for LEC Panx1 in atherosclerosis. New findings reveal variations in serum lipid levels, plaque size, and plaque vulnerability based on sex. However, these differences are not linked to the presence or absence of Panx1. Given the data presented, the significance of the results remains uncertain.

6. PLOS authors have the option to publish the peer review history of their article (what does this mean?). If published, this will include your full peer review and any attached files.

Reviewer #1: **Yes: **Takuro Miyazaki

Reviewer #2: No

---

## [Author Response · Author response to Decision Letter 0]

18 Nov 2024

Reviewer 1

We thank the Reviewer for his/her critical comments that helped us to improve our manuscript. We are happy to note that the Reviewer considers our results as clear without contradictions and that the novelty of our study was evident. We have responded to each of his/her remarks in detail below. 

Specific comments:

1) The authors conclude that the promotion of atherosclerosis in female Prox1-CreERT2+Panx1 fl/flApoe-/- mice affects T cell accumulation in the lesions, yet this study does not assess T cell accumulation in the lesions or the populations of T cells in the blood. Since T cells are roughly classified into proinflammatory and anti-inflammatory subsets, both quantitative and qualitative evaluations are essential. Previous reports (PMID: 36519468) have indicated that dysfunction of lymphatic endothelial cells reduces the stability of anti-inflammatory regulatory T cells systemically, increasing susceptibility to atherosclerosis, suggesting that impaired cellular transport via lymphatics may not always be the primary cause. In the author’s context, evaluating lymphatic endothelial cell functions, such as adhesion molecules, is warranted.

Response: We thank the Reviewer for this insightful suggestion, which helped us to further understand the potential mechanism underlying the sex-specific increased progression of atherosclerosis in female Panx1LECdelApoe-/- mice. Following the suggestion of the Reviewer we show in the revised manuscript that 1) as compared to Panx1fl/flApoe-/- controls, the number of T lymphocytes is increased in advanced plaques of female Panx1LECdelApoe-/- mice but not in male mice of the same genotype (Figures 8E and 10E), and 2) the expression level of VCAM-1 was lower in Panx1-deficient LECs than in Panx1-expressing LECs isolated from lymph nodes of female mice (Figure 10F). Although we appreciate the suggestion to classify the populations of T lymphocytes in the blood, this requires fresh samples and thus new studies with 10 weeks high cholesterol diet on all groups of mice (40 mice), which was not possible in the 6 weeks given for the revision of our manuscript.

2) The authors should discuss why the observed sex differences occur, in terms of Panx1. The current draft provides extensive explanations regarding gender differences in dyslipidemia; however, as these do not lead to a definitive conclusion, they should be condensed. Instead, the discussion should focus on why there are gender differences in Panx1's contribution to atherosclerosis. It would be beneficial to examine the differences in Panx1 expression between males and females.

Response: The new results mentioned under the above point 1 are extensively discussed in the revised manuscript (page 20, lines 1-20), resulting in a better balance between the discussion on sex-differences in early atherosclerosis that are independent of Panx1 and the sex-differences in the progression of atherosclerosis, which involves Panx1 in LECs. As requested, we mention in the revised manuscript that Panx1 expression is higher in LECs of males (page 17, line 1), an observation that was already described in detail in our earlier study (Ehrlich et al., Physiol Rep 2024; reference 18 in this manuscript).

3) Lipoprotein cholesterol and triglycerides should be fractionated into at least VLDL, LDL, and HDL for evaluation. Merely assessing changes in total cholesterol is insufficient to rule out the involvement of dyslipidemia.

Response: As requested by the Reviewer, we have performed additional measurements and show now the serum LDL and HDL values (Figures 3C-D, 7C-D and 9C-D). Unfortunately, VLDL measurements are not possible on the Cobas C111 analyser available to us.

4) The two experimental strategies (initiation and progression) are not clearly linked to the figures, even in the legends. Please revise this for clarity.

Response: We thank the Reviewer for this suggestion. We revised the legends and the two experimental strategies are now clearly mentioned for all Figures (pages 23-25). In addition, we have critically revised the text of the manuscript and the title of the manuscript to better link results and strategies. 

Reviewer 2

We thank the Reviewer for his/her comments on our manuscript.

The goal of this manuscript is to examine the impact of lymphatic endothelial Pannexin 1 (LEC Panx1) on the development and stability of atherosclerotic plaques. Despite being an intriguing subject, the data provided does not indicate a role for LEC Panx1 in atherosclerosis. New findings reveal variations in serum lipid levels, plaque size, and plaque vulnerability based on sex. However, these differences are not linked to the presence or absence of Panx1. Given the data presented, the significance of the results remains uncertain.

Response: We are happy to note that the Reviewer considers the subject of our study intriguing. Following the suggestions of Reviewer 1, we performed additional experiments to better understand the potential mechanism underlying the sex-specific increased progression of atherosclerosis female Panx1LECdelApoe-/- mice. We show in the revised manuscript that 1) as compared to Panx1fl/flApoe-/- controls, the number of T lymphocytes is increased in advanced plaques of female Panx1LECdelApoe-/- mice but not in male mice of the same genotype (Figures 8E and 10E), and 2) the expression level of VCAM-1 was lower in Panx1-deficient LECs than in Panx1-expressing LECs isolated from lymph nodes of female mice (Figure 10F). These new results mentioned are extensively discussed in the revised manuscript (page 20, lines 1-20), resulting in a better balance between the discussion on sex-differences in early atherosclerosis that are independent of Panx1 and the sex-differences in the progression of atherosclerosis, which involves Panx1 in LECs.

Journal Requirements

https://journals.plos.org/plosone/s/file?id=wjVg/PLOSOne_formatting_sample_main_body.pdf and https://journals.plos.org/plosone/s/file?

Response: The manuscript has been formatted according to the template.

To comply with PLOS ONE submissions requirements, in your Methods section, please provide additional information regarding the experiments involving animals and ensure you have included details on (1) methods of sacrifice, (2) methods of anesthesia and/or analgesia, and (3) efforts to alleviate suffering.

Response: All required details are in the methods section (page 7).

When completing the data availability statement of the submission form, you indicated that you will make your data available on acceptance. We strongly recommend all authors decide on a data sharing plan before acceptance, as the process can be lengthy and hold up publication timelines. Please note that, though access restrictions are acceptable now, your entire data will need to be made freely accessible if your manuscript is accepted for publication. This policy applies to all data except where public deposition would breach compliance with the protocol approved by your research ethics board. If you are unable to adhere to our open data policy, please kindly revise your statement to explain your reasoning and we will seek the editor's input on an exemption. Please be assured that, once you have provided your new statement, the assessment of your exemption will not hold up the peer review process.

Response: Once YARETA DOI has been finalized to make it publicly available, no further changes can be made. It is therefore advised to only do this for accepted manuscripts. We will do so immediately after acceptance of the manuscript and provide the DOI in the proofs of the manuscript.

 This study was supported by grants from the Swiss National Science Foundation (grant number 310030_182573 to BRK) and the Gabbiani fund (to AE). 

Response: This statement has been included in the revised manuscript (page 22). 

NO authors have competing interests. 

Response: The above requested changes have been made (page 22).

---

## [Decision Letter · Decision Letter 1]

27 Nov 2024

Protective role of Pannexin1 in lymphatic endothelial cells in the progression of atherosclerosis in female mice

PONE-D-24-40523R1

Dear Dr. Kwak,

We’re pleased to inform you that your manuscript has been judged scientifically suitable for publication and will be formally accepted for publication once it meets all outstanding technical requirements.

Kind regards,

Eliseo A Eugenin, Ph.D.

Academic Editor

PLOS ONE

Additional Editor Comments (optional):

Dear Dr. Kwak:

Thank you for re-submitting your manuscript to PLOsone. The manuscript was evaluated by 2 experts in the field with excellent comments. Thank you for addressing all the comments.

Best Regards

Eliseo Eugenin

Reviewers' comments:

Reviewer's Responses to Questions

**Comments to the Author**

1. If the authors have adequately addressed your comments raised in a previous round of review and you feel that this manuscript is now acceptable for publication, you may indicate that here to bypass the “Comments to the Author” section, enter your conflict of interest statement in the “Confidential to Editor” section, and submit your "Accept" recommendation.

Reviewer #1: All comments have been addressed

Reviewer #2: All comments have been addressed

2. Is the manuscript technically sound, and do the data support the conclusions?

Reviewer #1: Yes

Reviewer #2: Yes

3. Has the statistical analysis been performed appropriately and rigorously? 

Reviewer #1: Yes

Reviewer #2: I Don't Know

4. Have the authors made all data underlying the findings in their manuscript fully available?

Reviewer #1: Yes

Reviewer #2: Yes

5. Is the manuscript presented in an intelligible fashion and written in standard English?

Reviewer #1: Yes

Reviewer #2: Yes

6. Review Comments to the Author

Reviewer #1: The authors have adequately addressed the concerns raised by this reviewer. There are no additional comments.

Reviewer #2: The authors have attempted to address my comments. The manuscript is written in an easy to understand manner.

7. PLOS authors have the option to publish the peer review history of their article (what does this mean?). If published, this will include your full peer review and any attached files.

Reviewer #1: **Yes: **Takuro Miyazaki

Reviewer #2: No

---

## [Editor Report · Acceptance letter]

13 Dec 2024

PONE-D-24-40523R1 

PLOS ONE

Dear Dr. Kwak, 

I'm pleased to inform you that your manuscript has been deemed suitable for publication in PLOS ONE. Congratulations! Your manuscript is now being handed over to our production team.

Kind regards, 

on behalf of

Dr. Eliseo A Eugenin 

Academic Editor

PLOS ONE